# Ion-Imprinted Polymers: Synthesis, Characterization, and Adsorption of Radionuclides

**DOI:** 10.3390/ma14051083

**Published:** 2021-02-26

**Authors:** Vipul Vilas Kusumkar, Michal Galamboš, Eva Viglašová, Martin Daňo, Jana Šmelková

**Affiliations:** 1Department of Nuclear Chemistry, Faculty of Natural Sciences, Comenius University in Bratislava, Mlynska dolina Ilkovicova 6, 842 15 Bratislava, Slovakia; eva.viglasova@uniba.sk; 2Department of Nuclear Chemistry, Faculty of Nuclear Sciences and Physical Engineering, Czech Technical University in Prague, Brehová 7, 115 19 Prague, Czech Republic; martin.dano@fjfi.cvut.cz; 3Department of Administrative Law and Environmental Law, Faculty of Law, Comenius University in Bratislava, Safarikovo namestie 6, 810 00 Bratislava, Slovakia; jana.smelkova@uniba.sk

**Keywords:** ion-imprinted polymers, radioactive waste, radionuclides, adsorption, separation

## Abstract

Growing concern over the hazardous effect of radionuclides on the environment is driving research on mitigation and deposition strategies for radioactive waste management. Currently, there are many techniques used for radionuclides separation from the environment such as ion exchange, solvent extraction, chemical precipitation and adsorption. Adsorbents are the leading area of research and many useful materials are being discovered in this category of radionuclide ion separation. The adsorption technologies lack the ability of selective removal of metal ions from solution. This drawback is eliminated by the use of ion-imprinted polymers, these materials having targeted binding sites for specific ions in the media. In this review article, we present recently published literature about the use of ion-imprinted polymers for the adsorption of 10 important hazardous radionuclides—U, Th, Cs, Sr, Ce, Tc, La, Cr, Ni, Co—found in the nuclear fuel cycle.

## 1. Introduction

Spent nuclear fuel (SNF) and high-level radioactive waste (HLRW) are generated during nuclear power reactor operation and decommissioning of nuclear facilities, respectively. There are 62 different radionuclides (RNs) formed from ²³⁵U fission, which due to their instability, are subject to radioactive transformation. About 200 different radioisotopes are observed during the nuclear reactor operation, mainly in the core. Nuclear power plants around the world are the major producers of SNF and HLRW. Situated on the territory of the Slovak republic are in Jaslovské Bohunice (Bohunice 4 and 3—reactor units in operation; Bohunice 2, 1 and A1—in the decommissioning process) and Mochovce (Mochovce 1 and 2—reactor units in operation; Mochovce 3 and 4—reactors units under construction).

Slovak legislation on SNF and radioactive waste (RW) management fully corresponds to international legislation and regulations. Requirements for the management of SNF and RW have been regulated in the Act No. 541/2004 Collection on the peaceful use of nuclear energy (Atomic Act) and in its implementing regulation and the Act No. 308/2018 Collection on the National Nuclear Fund and amending the Atomic Act. The Atomic Act defines RW as any unusable material in gaseous, liquid, or solid form, which cannot be released into the environment due to the content of RNs or the level of RN contamination. Details on the requirements for the management of nuclear materials, RW, and SNF are also defined by the Decree of the Nuclear Regulatory Authority of the Slovak Republic No. 30/2012 Collection. It is assumed that the individual units of nuclear power plants in Slovakia will produce 2500 tons of SNF and 3700 tons of HLRW during their operation period. Produced SNF can be considered either as a usable source of energy and RNs and can be reprocessed or can be destined for final disposal if it is considered as RW.

By ²³⁵U fission two main groups of RNs are created:(1)*Fission products (FPs)*: formed by the fission of fuel nuclei by neutrons. These are highly active RNs; whose half-life ranges from a few seconds to thousands of years:
-volatile substances (⁹⁹Tc, ¹⁰³^,^¹⁰⁶Ru, ¹³¹^,^¹³³^,^¹³⁵I, ¹³⁴^,^¹³⁵^,^¹³⁷Cs, …),-non-volatile substances (⁹⁰Sr, ¹⁴⁰La, ¹⁴¹^,^¹⁴⁴Ce, …),-noble gases (⁸⁵Kr, ¹³¹^,^¹³³^,^¹³⁵Xe), tritium (³H).
(2)*Activation products (APs)*: formed by the interactions of neutrons with inactive nuclides of the coolant, moderator, or reactor construction materials, respectively. By activating the reactor construction materials, the following main activation corrosion and erosion products are formed:
-steel: ⁵¹Cr, ⁵⁴Mn, ⁶⁰Co, ⁶³Ni, ⁵⁹Fe, ⁹³Mo, ⁹⁴Nb, ¹⁰⁸ᵐ^,^¹¹⁰ᵐAg, ¹²⁴^,^¹²⁵Sb, ¹⁵²Eu, ¹⁶⁶ᵐHo,-concrete: ³⁶Cl, ⁴¹Ca, ⁴⁶Sc, ⁵¹Cr, ⁵⁹Fe, ⁵⁹, ⁶³Ni, ⁶⁰Co, ⁶⁵Zn, ⁸⁵Sr, ¹²⁴Sb, ¹³¹Cs, ¹⁵²^,^¹⁵⁴^,^¹⁵⁵Eu, ¹⁶⁰Tb, ¹⁸¹Hf, ¹⁸²Ta.

The gradual capture of neutrons by fuel nuclei and their subsequent radioactive transformations produce transuranic elements. The most commonly used source nuclides of transuranic elements are ²³⁸U, ²³⁷^,^²³⁹Np, ²³⁸^,^²⁴²Pu, ²⁴¹^,^²⁴³Am, ²⁴²^,^²⁴⁴Cm. The RNs with the half-life longer than >200,000 years, such as: ⁷⁹Se, ⁹³Zr, ⁹⁹Tc, ¹⁰⁷Pd, ¹²⁶Sn, ¹²⁹I, ¹³⁵Cs which are difficult to measure, play significant roles in the separation processes. However, there are also other problematically detected RNs with shorter half-life < 200,000 are i.e., ⁵⁹^,^⁶³Ni, ⁷⁹Sc, ¹⁴⁶Sm, and actinide ²³⁶U.

*Uranium* is the most significant metal for energy production and also the primary waste generated from nuclear reactor operation. Uranium is known for its toxicity, apart from the fact that ingestion of uranium can damage the liver and cause acute renal failure in humans [1]. The uranyl ion (UO₂⁺) occurs in oxidation state 6+ and is the most stable form of uranium in aqueous solution [2,3,4]. According to safety regulations, uranium consumption in drinking water should be less than 0.015 mg/L [5]. *Thorium* is a metallic element of the actinides. For energy generation, thorium is a better choice than uranium due to its abundance on Earth, and the relatively less radioactive waste generated from its operation in a nuclear reactor. More than 99.99% of natural thorium is ²³²Th, the rest being represented by ²³⁰Th and ²²⁸Th. Thorium metal occurs naturally in the environment. The increasing release of these metals has spiked due to mining activity, e.g., milling, and processing operation, coal production, and phosphate production for fertilizers. Besides its attractive properties for nuclear energy generation, it is toxic element for human health, as a potentially carcinogenic substance. Thorium exposure is primarily due to inhalation, intravenous injection, ingestion and absorption through the skin [6]. The dispersion of thorium in the environment, mainly in water bodies, creates great concern to humans and ecological life forms [7,8]. *Cesium-137* and *Strontium-90* are isotopes produced during a nuclear power plant’s operation by ²³⁵U fission and are the most harmful RNs due to their strong beta emission properties and long half-life (30 years, and 29 years, respectively). The nuclear accidents in Chernobyl and the Fukushima Daiichi released large amounts of cesium isotopes (¹³⁴Cs, ¹³⁷Cs) into the surroundings [9]. These radionuclides are highly soluble and mobile in water. ⁹⁰Sr can accumulate in human organs, mostly in the liver, lungs, and kidneys of animals, and causes diseases and damage [10,11]. *Cerium-144* (half-life 284.91 days) is formed by the fission of uranium and accumulates in the human body leading to acute myocardial infarction, leukemia, and imbalance in blood biochemistry. It also causes a toxic effect damaging the cell membranes in marine life, consequently damaging the nervous system and reproduction system in organisms [12,13]. The nuclear power generation from uranium generates a significant amount (≈6.05%) of *Technetium-99*, which as a FP is highly toxic due to its high mobility in water and a half-life of 2.11 × 10^5^ years [14]. *Lanthanum-140* (half-life 1.68 days) occurs as a FP of uranium; it has no biological role and is considered an environmental threat due to its radioactivity [15]. *Chromium-51* (half-life 27.70 days) is an AP and can be used as a radioactive tracer in isotope dilution analysis. Chromium is needed in trace amounts for the proper functioning of the human body, but its excess can cause carcinogenic effects [16]. *Nickel-59, 63* are isotopes generated in the nuclear reactor as APs [17]. ⁵⁹Ni and ⁶³Ni have a half-life of 76,000 and 101.2 years, respectively [18]. Nickel is responsible for various health effects on humans, depending on the route of exposure and dosage. It can cause cardiovascular diseases, skin diseases, nose and lung cancer [19]. *Cobalt-60* is generated in the nuclear power plant. ⁶⁰Co is the source of high gamma radiation and has a half-life of 5.27 years. The presence of cobalt in the environment, considering its radioactivity, can cause detrimental effects on ecology and human health [20].

For the mitigation of radionuclides’ environmental impact, various methods have been used to date such as phytoremediation [21], ion-exchange [22], chemical precipitation [23], solvent extraction [24], reverse osmosis [25], electrochemical purification [26], membrane separation [27] and adsorption [28]. The adsorption method is the most reliable method for toxic radionuclide removal and has been widely used. Numerous materials can be utilized for adsorption purposes such as inorganic sorbents [29], oxides [29,30,31], carbon materials [32], organic-inorganic hybrid materials [33], metal-organic frameworks [34] and porous organic polymers [35]. Although the adsorbents used in nuclear waste management, e.g., ion exchangers and inorganic materials are constantly improving over the time in their structural, physical, and chemical aspects [36], these materials also have disadvantages, e.g., rapid saturation, etc. 

The Department of Nuclear Chemistry, Faculty of Natural Sciences, Comenius University in Bratislava, has long been dedicated to the development and research on various highly selective and radiation-stable sorbents, which are used:-as a component in a multibarrier system for the safe storage and final disposal of SNFs and RWs, respectively,-as a separation component for RN isolation from RN mixtures, mainly for further use, or in radiochemical analysis for target radionuclide separation and subsequent qualitative and quantitative determination.

Ion-imprinted polymers (IIPs) can solve this issue and are widely used for in radionuclide adsorption or removal processes [37]. IIPs are sorbents with various application possibilities, e.g., extraction, filtration, solid and liquid media purification [38]. These materials are crosslinked polymers with pores and binding sites for the targeted ions and negatively or positively charged molecules [39]. IIPs are generally prepared from a reaction mixture composed of a functional monomer, a crosslinker, an initiator and a template [40]. Imprinting of the template ions in a polymer matrix can be achieved by using different strategies and are discussed further in detail. IIPs are developed in a similar way to imitate key and lock mechanisms (Figure 1) for recognition and removal of the targeted ions. Thus, IIPs offer excellent selectivity and specific affinity for a given ion. Contaminants present at low concentrations can be removed selectively by IIPs, which was not effectively achieved before by other methods [41]. IIPs are similar to molecularly imprinted polymers (MIPs) which are developed to imitate the interactions between enzymes and antibodies. The difference between IIPs and MIPs lies in the type of recognition substance which can be ions or molecules.

MIP and IIP preparation differ in the type of interaction between the monomers or ligands (MIPs) and ions (IIPs), i.e., covalent or noncovalent interactions. The covalent molecular imprinting technique method developed by Wulff et al. [42], involves reversible covalent binding between monomers and the templates. This approach provides greater stability and consistent recognition sites compared to the non-covalent approach. In the non-covalent method, the non-covalent bonds such as H-bonds, ionic bonds, van der Waals interactions, etc. are used by monomers to self-assemble around the template. The template is then removed from the material after polymerization. Mosbach et al. [43] developed the non-covalent method, which is an effective way to form cavities with the template’s shape and charge memory effects, responsible for the selective recognition of analytes. The non-covalent method is easy to use and has the option to choose from a range of different monomers. IIPs exhibit similar virtues as MIPs, the only difference being the specific recognition of inorganic ions after the imprinting process [44,45]. The first IIP preparation was achieved using poly(vinyl pyridine) crosslinked with 1,4-dibromobutane in the presence of metal ions [46]. Saunders et al. [47] used 2-chloroacrylic acid and ethylene glycol dimethacrylate (EGDMA) with a uranyl ion-imprinted co-polymer after removal of the template. This material selectively extracts uranium from dilute aqueous solutions. He et al. [48] synthesized a new functional monomer—*N*-(*o*-carboxyphenyl) maleamic acid (CPMA)—for Th(IV) ion separation. This adsorbent was made of a silica gel surface coating with a thin IIP layer. 

A few reviews can be found on IIP technologies for adsorption/removal of a larger group of metal ions such as a review on IIPs for removing inorganic pollutants by Mafu et al. [49]. Further, Hande et al. [50] described synthesis methodologies for several applications, and recent advances in imprinted polymers can be found in the publication of Branger et al. [51]. A review article on imprinted polymers developed from halloysite nanotubes for environmental pollutant removal can be found [52] and more discoveries were published in [53,54]. To the best of our knowledge, there is no specific review describing the use of IIPs for specific radionuclides. Our study’s focus is to provide an update on the theoretical background about the adsorption of ten important radionuclides—U, Th, Cs, Sr, Ce, Tc, La, Cr, Ni, Co—by IIPs and its fundamental aspects. In the following sections, the fundamentals of IIPs and IIPs’ adsorption processes of various radionuclides are discussed.

## 2. Fundamentals of IIPs Synthesis for Radionuclides Adsorption

The synthesis of IIPs is commonly carried out in the following way: First, monomers containing functional groups are mixed with template ions, whereupon the monomers self-assemble around the ions. Second, a crosslinker is used to polymerize the monomers by using a photo- or thermal polymerization technique. Third, template ions are removed from the polymers, consequently generating specific binding sites that can capture target ionic species [55]. IIPs shows excellent ion selectivity due to their recognizable binding sites for a particular ion’s size and charge. The adsorption capacity of the IIPs is influenced by certain factors, such as their ligands’ ability to bind with metal ions, ionic charge, the size of ions, and the electronic configuration of the metals such as coordination number or oxidation states [56,57]. The IIPs are stable against pH, temperature, and pressure which is almost impossible to achieve in natural molecular recognition systems [58].

### 2.1. Principles and Basic Components for the Synthesis of IIPs

One of the most crucial criteria for the effective binding and selectivity of the metal ion to the imprinted material is the interaction of the functional groups of monomers with the template [59]. The template interaction depends upon the type of bonding present with monomers. Two approaches, as mentioned earlier, are used: the covalent approach and the non-covalent approach. The IIPs are developed using a template, functional monomers, crosslinkers, porogens, and initiators. The ratio and the selection of these components are essential for better selectivity and binding capacity and affect the ultimate physical and chemical properties of the IIPs.

#### 2.1.1. Templates

The selection of templates is a crucial requirement for IIP development. The templates’ chemical and physical properties are analyzed for their selection. The basic requirements for choosing a template are as follows:The template should be inert, it should not affect the cross-linking and polymerization of the monomer, e.g., the functional moieties shouldn’t interfere with the polymerization by acting as inhibitors or reacting with the monomer.The template should be cost-effective.The template should be stable at the polymerization or crosslinking temperature or exposure to UV radiation.

#### 2.1.2. Monomers and Crosslinkers

In order to prepare recognition sites in a polymer matrix monomer should have functional groups that will bind with the analytes. Monomer selection is an important factor for the design of IIPs. To prepare imprinted polymers a free radical polymerization (FRP) method is used since it is easy to execute and offers a wide range of monomer selection. Generally, in radical polymerization (RP) of IIPs vinyl monomers are used, e.g., methacrylic acid [60], acrylamide [61], styrene [62], 4-vinylpyridine [63], etc. The monomers are crosslinked using di-, tri-, or tetrafunctional vinylated crosslinkers. Crosslinkers are useful for improving the mechanical properties of the material such as strength and tolerance to the solvent and different pH values. They also help the morphology of the material and adjusting the monomer–crosslinker ratio affects the materials’ porosity. Commonly used crosslinkers for free radical polymerization are ethylene glycol dimethacrylate (EGDMA) [64], divinylbenzene (DVB) [65], or trimethylolpropane trimethacrylate (TRIM) [66]. The ratio of the template to monomer is crucial. The compatibility ratio for the adsorption of UO₂⁺ ion was found to be 1:4 template to monomer, and exceeding the ratio will cause reduced interaction with the template [67]. The abovementioned monomers’ binding capacity is relatively weaker; they carry a single functionality for interaction. The dependence on a metal ion’s characteristics such as shape, oxidation, and chemical structure and surroundings play an important role in its effective binding [68]. To improve the binding performance of the materials, various ligands were prepared for effective binding with selective ions. Fasihi et al. [69] synthesized a 1-hydroxy-2-(prop-2′-enyl)-9,10-anthraquinone ligand to prepare uranyl IIPs. Zulfikar et al. [70] prepared 5,7-dichloroquinoline-8-ol and 4-vinylpyridine-based IIPs for the adsorption of yttrium. Macromolecular monomers such as chitosan and, sodium alginate are also widely used to remove radionuclides from the environment [71,72]. Chitosan has NH₂ and -OH groups that will donate to the metals and be useful for chelation and crosslinking [73]. Sodium alginate contains -COOH and -OH groups, which helps provide the necessary binding sites for the preparation of IIPs [74]. 

#### 2.1.3. Porogens (Solvents) 

The purpose of the solvent in the polymerization is to solubilize the components of the reaction mixture such as monomers, templates, and crosslinkers. It helps in pore formation in a material, which improves the material’s adsorption properties [75]. The solvent choice depends on the reaction system interactions. In the case the system consists of organic molecules that contain hydrogen atoms, for that purpose nonpolar solvents such as toluene can be used, and for polar systems, water or other polar solvents have been used [76].

#### 2.1.4. Initiators

The initiator is responsible for the activation of free radical polymerization in monomers. The initiator selection depends on the template ion’s stability and electrostatic interactions with the monomers, mainly for complex formation. If initiators are thermally activated, thermal stability of the template is also required. If the monomers form a complex with the template ions by hydrogen bond formation, then it is not appropriate to use a thermal method. However, a photochemically activated initiator is more reliable for using such a low stability template and its interactions. The initiator concentration is directly related to the generation of radicals in the system, which increases the particle size of the IIPs [77]. Some of the initiators used are shown in Figure 2.

### 2.2. Imprinting Strategy for the IIPs Synthesis 

The preparation of IIPs can be performed in several ways using multiple preparation methods. In this section, the most commonly used type of polymerization or imprinting strategies are described. The simplest method for IIP synthesis is radical polymerization (RP). Radical polymerization can be divided into bulk polymerization [78], emulsion polymerization [79], solution polymerization [80], precipitation polymerization [81] and reversible addition-fragmentation techniques (RAFT) polymerization [82]. In the living free radical polymerization method (for e.g., RAFT) the desired polymer morphologies can be achieved [83]. RP also provides significantly faster polymerization rates compared to the condensation polymerization method. The system in RP contains the monomer, initiator and crosslinker, if required. Another common method for making crosslinked structures is the condensation method, which includes sol-gel and co-condensation polymerization. Chitosan is a functional group-carrying polymer with -NH_2_ and -OH groups which can be modified with other compounds, in order to form a dendritic structure or crosslinked networks. For the preparation of crosslinked structures chitosan can be crosslinked with various difunctional molecules such as glutaraldehyde [84], epichlorohydrin, etc. In another study, Gao et al. [85] used 3-glycidoxypropyltrimethoxysilane (GPTMS) for crosslinking chitosan or silane coupling agents. Imprinting metal ions on the materials’ surface solves certain drawbacks associated with the use of IIPs, such as poor mass transfer and binding kinetics. The surface of imprinted polymers can improve the selectivity and specificity of the materials’ surface, which is not found in general IIPs [86].

## 3. Adsorption Performance of IIPs

### 3.1. Quantities for Measuring Adsorption Performance of IIPs

IIPs are known for their selectivity towards the analytes, and IIPs’ adsorption performance can be studied with respect to a non-imprinted polymer (NIP) which is synthesized using the same procedure but without the template ion. Two different equations are used for calculation, one is the % *removal* and the other one is equilibrium adsorbate concentration. The *% removal* represents the approximate value of the adsorption performance and it is shown in Equation (1):*% removal* = *100**×**(C₀ − C_e_)/C₀*(1)
where *C_o_* (mg/L) is the initial adsorbate concentration and *C_e_* (mg/L) equilibrium concentration in the solution. The equation for the adsorption performance (Equation (2)) is derived from the expression of the equilibrium concentration of the adsorbate *Q_e_* (mg/g) at the amount of adsorbate adsorbed at the equilibrium [87]:*Q_e_* = *V ×**(C₀ − C_e_)/m*(2)
where *m* (g) is the dry mass of the used adsorbent; *V* (L) is the volume of the adsorbate solution.

### 3.2. Adsorption Isotherm

Adsorption isotherms can be defined by the relationship between the amount of adsorbate adsorbed on the adsorbent at equilibrium and the adsorbate’s concentration at equilibrium in the solution. This relationship is determined by plotting *Q_e_* against *C_e_*. The Data modeling is achieved using different models such as Langmuir, Freundlich, etc. [88]. To understand the binding capacity of IIPs, the graphs are compared with non-imprinted polymers (NIP). The Langmuir model only deals with a measurable number of the binding sites on an IIP’s surface and is also limited to monolayer adsorption [89]. In IIPs, the Langmuir model was used based on two assumptions: the adsorbed molecules will not interact, and after the analyte occupies the available binding sites, no further adsorption will be possible [90]. A linear form of the Langmuir Equation (3) is presented as follows:*1/q_e_* = *(1/q_max_* × *b* × *C_e_)* + *(1/n)* × *log C_e_*(3)
where *q_e_* is the equilibrium of the amount of uranyl ions adsorbed on the adsorbent (μmol/g), *q_max_* is the maximum adsorption capacity (μmol/g), *C_e_* is the equilibrium concentration of uranyl ions in the solution (μmol/L), *b* is the Langmuir constant (L/μmol). *q_max_* and b are the Langmuir constants, which can be calculated from the intercept and slope of the linear plot based on *1/q_e_* versus *1/C_e_*. 

The Freundlich model can be applied on heterogeneous surfaces with the extension of multilayer adsorption [58]. Similarly, the linear Freundlich isotherm equation can be presented as follows:*log q_e_* = *log K_f_* + *(1/n)* × *log C_e_*(4)
where the *K_f_* denotes the Freundlich isotherm constant and *1/n* is the heterogeneity factor.

The Langmuir and Freundlich isotherms are commonly used isotherms in order to describe sorption experiments. However, other different isotherms models are also studied instead of the Langmuir or Freundlich isotherms. To investigate the adsorption of Cs⁺ ion on two different RAFT IIPs, Meng et al. [91] applied three isotherm models: Freundlich, Langmuir, and Redlich-Peterson. In this study, the Redlich-Peterson model was found to be more relevant to the experimental data, based on correlation coefficient values. In another study, apart from the Langmuir and Freundlich isotherm model, the Dubinin-Radushkevitch (D-R) model was successfully applied and gave a better fit for the adsorption of uranium in comparison with previous ones. Sadeghi et al. [87] applied both Freundlich and Langmuir models for finding the binding capacity of an IIP, and the experimental results showed that Langmuir isotherm better corresponded with the experimental data, based on correlation factors. 

### 3.3. Influence of pH onto Radionuclides Adsorption by IIPs

Changes in the solution acidity or basicity affect the concentration of H⁺ ions in a solution. At lower pH values, the self-assembly process reduces the choice of metal ions over H⁺ ions. Thus, a higher pH value favors complex formation. On the other hand, the pH also determines the material’s surface properties and protonation level, which directly control the electrostatic interactions with the interacting species. In the study carried out by Sadeghi et al. [92], experiments performed in a strongly acidic medium revealed that uranyl ions’ removal capacity decreased due to the protonation of the amine group in the IIP adsorbent. Monier et al. [69] found that acidic conditions bind with the active amidoxime sites in the adsorbent, which leads to a lower adsorption capacity, whereas at values up to pH~5, the UO₂⁺ ion removal efficiency is increased. In the range of pH values higher than pH~5, uranyl ions are precipitated in the form of UO₂OH⁺ and UO₂(OH)₂. These hydrolyzed forms caused a decrease in the adsorption performance [72]. pH~6 was found to be optimal for strontium and the increase of pH caused less the adsorption efficiency due to hydrolyzed ion species [93]. The lower pH value forms carboxylate [79] and amino groups [94] in IIPs, which leads to a decrease in adsorption capacity. A similar adsorption behavior was found in the case of other RNs, e.g., cobalt [95] and technetium [96].

### 3.4. Influence of Dosage and Concentration onto Radionuclides Adsorption by IIPs

Except for the pH, other essential factors for optimal adsorption are the adsorbent dosage and adsorbate concentration. Tawengna et al. showed an increase in extraction efficiency from 47 to 80% and 35 to 62% with an increase in the amount of adsorbent from 10 to 50 mg for magnetic IIPs and NIPs, respectively. This increased extraction efficiency is explained due to the rise in adsorption sites and surface area for analyte adsorption. The initial concentration of the uranyl ion solution was varied from 0.5 and 8 mg/L using 50 mg of IIP adsorbent, and the experiments showed that the saturation point was reached at 2 mg/L of initial concentration. Maximum adsorption capacities of 1.04 ± 0.03 and 0.95 ± 0.02 mg/g were observed for magnetic IIPs and NIPs, respectively. This adsorption capacity is increased due to the increased strength of the transfer force and the mass transfer [90]. In another study, an adsorbent dosage in the range of 10–100 mg was applied for uranyl ion extraction from an aqueous solution. The study proved that the maximum extraction efficiency was achieved at 50 mg, and a further increase of adsorbent dose did not improve the performance of the adsorbent [97].

## 4. Application of IIPs for Radionuclide Removal

### 4.1. Uranium

Uranium occurs in Nature mainly in the forms of 4+ and 6+ oxidation states. The most common uranium form is uranium oxide, the uranyl cation UO₂⁺, and it is the most generally used template for IIP production [98]. Ahmadi et al. [99] prepared uranyl ion-imprinted polymers applying the following steps: (a) formation of a uranyl binary complex with N,N-ethylenebis(pyridoxylideneiminato); (b) synthesis of a [UO₂(pyr₂en)DMSO]Cl₂ ternary complex with 4-vinylpyridine, and (c) polymerization of the ternary complex with styrene. The prepared polymer showed excellent selectivity and adsorption performance towards uranyl ion compared to other heavy metals. The use of quinoline-8-ol functionalized by 3-aminopropyltrimethoxysilane-modified silica nanoparticles (HQ-APTMS-SI) is mainly used for imprinted polymer nanosphere preparation. This was developed and studied primarily for selective uranyl ion capture for the decontamination of nuclear power plant effluents by Milia et al. [100]. This method is schematically shown in Figure 3. The interpenetration networks of crosslinked hydrogel uranyl IIPs were prepared from chitosan and polyvinyl alcohol with the difunctional crosslinker ethylene glycol diglycidyl ether developed by Liu et al. [101]. Zhou et al. [102] prepared magnetite uranyl IIP from chitosan crosslinked with glutaraldehyde. Sadeghi et al. [92] prepared uranyl-imprinted silica-coated magnetic nanoparticles. The resultant polymer is a base platform for the development of functionalized nanomaterials for imprinted materials.

Pakade et al. [98] prepared a novel uranyl ion-imprinted polymer from 1-(prop-2-en-1-yl)-4-(pyridin-2-ylmethyl) piperazine and methacrylic acid, crosslinked by using EGDMA. Monier et al. [76] prepared a cellulose-based imprinted polymer for uranyl ion removal. The cellulose was modified with salicylaldehyde, which was crosslinked in the presence of uranyl template with formaldehyde. Zhang et al. [99] developed a typical uranyl ion-imprinted polymer using free radical polymerization from the vinyl monomers 2,4-dioxopentan-3-yl methacrylate and the crosslinker EGDMA in the presence of uranyl ion as a template. Tavengwa et al. [100] developed the new uranyl ion magnetic imprinted polymer embedded with γ-methacryloxypropyltrimethoxysilane (γ-MPS). The silane monomer not only helps the polymer, but also helps to stabilize the Fe_3_O_4_ nanoparticles. In another study, Monier et al. [69] prepared amidoximated modified sodium alginate (Na-Alg) for the development of U-IIPs using glutaraldehyde as a crosslinker. The same authors prepared uranyl ion chelating microsphere using salicylaldehyde and *p*-aminostyrene (Schiff base), mainly used for the complex formation with UO₂⁺ ion and later used as crosslinked with divinylbenzene (DVB) [101]. Tavengwa et al. [85] prepared oleic acid-modified magnetic nanoparticles coated with the imprinted polymer by a precipitation-polymerization method. Meng et al. [57] synthesized amine-functionalized silica, later modified with acrolyl chloride. The prepared vinylated silica was crosslinked with EGDMA and MAA in the presence of uranyl ion. Tavengwa et al. [102] prepared 3 γ-(methacryloxy)propyltrimethoxysilane (γ-MPS) coated magnetic uranyl ion-imprinted polymer in the form of nanocomposite beads using salicylaldoxime (SALO) as 4-vinylpyridine (4-VP) monomer by a bulk polymerization technique. Yang et al. [103] developed a uranyl ion mesoporous silica functionalized using diethylphosphatoethyltriethoxysilane (DPTES) and poly(ethylene glycol)-block-poly(propylene glycol)-block-poly-(ethylene glycol) copolymer (P123). These materials have shown excellent radioresistance stability and high regeneration capacity in acidic and radioactive media and are shown in Figure 4.

Zhu et al. [104] used an interesting approach for generating ion-imprinted hierarchical porous carbon materials by using a hyper-accumulation method. The plant *Suaeda glauca* was used for this purpose to hyperaccumulate uranium ion from aqueous solution, later the material was kept for carbonization, mainly after successful ion removal from the carbonized material. The material was then used for further investigations. In another study, Wang et al. [105] prepared highly selective and sensitive uranium sensors. The imprinted polymer was produced using the sol-gel method where isophthalaldehyde-tertrapyrrole (IPTP) was used as a ligand for the synthesis, and α-methacrylic acid was used as a functional monomer. The uranyl ion template was used for the preparation, whereas the polymer was coated on a carbon paste electrode (CPE) (Figure 5).

In a recent study, Zhong et al. [106] used a simple method of preparing the U-IIPs by using hydrothermal crosslinking of chitosan in the presence of uranyl ion as a template. The adsorption performance and other details of the materials are provided in Table 1.

### 4.2. Thorium

Being the first true actinide, thorium still has an empty 5f orbital, and therefore Th(IV) is the most stable and plentiful oxidation state [111]. The IIPs are an excellent choice for Th⁴⁺ removal due to their selectivity and retention capacity. Lin et al. [112] prepared Th⁴⁺-IIP using a new pyrazole derivative—1-phenyl-3-methylthio-4-cyano-5-acrylicacidcarbamoylpyrazole—as a complexing monomer and by using carboxylic acid-functionalized silica modified with maleic anhydride in the presence Th⁴⁺ as a template. In another work of Lin et al. [78], methacrylic acid as a complexing agent and modified carboxylic acid-functionalized silica were used for Th-IIP preparation. Similarly, a modified silica with acryloyl chloride and dibenzoylmethane used as metal chelating agent was presented by Ji et al. [80]. The magnetic Th-IIPs prepared by He et al. [113] apply a novel complex—N,N′-bis(3-allyl salicylidene) *o*-phenylenediamine (BASPDA)—on the surface of Fe_3_O_4_-SiO_2_. Magnetic chitosan crosslinked with epichlorohydrin (ECH)-based Th-IIPs were prepared by Huang et al. [114] where the adsorption performance was investigated. Othman et al. [115] reported radiation-induced copolymerization of 2-hydroxyethyl methacrylic phosphoric acid diester (2-HMPAD) and 2-hydroxyethyl methacrylic phosphoric acid monoester (2-HMPAA)–thorium ion complex using as crosslinker DVB or EGDMA where the base represents a non-woven fabric made of polypropylene/polyethylene. The adsorption performance of IIPs mentioned above is presented in Table 2.

### 4.3. Cesium 

To capture the cesium from the environment, different materials based on carbon, titanate, tungstate, vanadate, hydroxyapatite, metal oxides were used [108,109,111,112,116,117]. IIPs based on these sorption materials were investigated in various studies. Zhang et al. [118] prepared sodium trititanate whiskers (STWs) supported by a chitosan ion-imprinted polymer for the selective capture of the Cs⁺ from aqueous solutions (Figure 6).

Shamsipur et al. [119] prepared Cs-dibenzo-24-crown-8 (DB24C8) chelate imprinted nanoparticles by a precipitation polymerization method. Iwasaki et al. [120] prepared polyacrylonitrile (PAN)-based Cs⁺-IIP for Cs adsorption from nuclear waste. Meng et al. [91] prepared two novel Cs⁺-IIPs by a surface imprinting technique supported by a SBA-15 mesoporous silica matrix. The polymerization was carried out using two different RAFT (Figure 7). Other important properties and details of the aforementioned IIPs are presented in Table 3.

### 4.4. Strontium

⁹⁰Sr is produced in the nuclear fission process of ²³⁵U and from nuclear weapons testing. In the environment it occurs in the 2+ oxidation state [121]. For Sr²⁺ removal, Liu et al. [122] used PTW and chitosan for IIP preparation. The PTW has mechanical and chemical stability and excellent wear-resistance properties. The chitosan was grafted on the surface of the PTW using a GPTMS silane coupling agent. Li et al. [11] have prepared strontium ion-imprinted hybrid gels from bis(trimethoxysilylpropyl) amine (TSPA) for the removal of Sr²⁺ and Ca²⁺ ions using dicyclohexano-18-crown-6, methacrylic acid (MAA), EGDMA removal. For the determination of Sr²⁺ in urine samples, Baharin et al. [123] prepared Sr²⁺-IIP by using dicyclohexano-18-crown-6 (D18C6), MAA, and EGDMA as the crosslinker. Palygorskite offers promising properties such as high surface area, mechanical and chemical stability. Pan et al. [124] prepared palygorskite and chitosan-based Sr-IIP hollow spherical beads. Microorganisms such as yeast formed the support for the preparation Sr-IIP used by Song et al. [94]. In this preparation, a composite silica/yeast was generated by the Stober method. The prepared composite was crosslinked with chitosan with KH-560. Liu et al. [125] developed a thermally responsive IIP based on SBA-15 mesoporous silica loaded with magnetic polyethyleneimine. The free-radical polymerization of vinyl monomers methacryloxypropyltrimethoxysilane (MEMO), *N-*isopropylacrylamide (NIPAM) and *N,N*-methylenebisacrylamide (MBAA) were used, which were polymerized onto the surface of the prepared silica.

Research on graphene and graphene oxide-based materials is currently at the forefront due to graphene’s properties such as 2D dimensional layered structure, mechanical, thermal, and chemical properties. Liu et al. [82] prepared a novel hydrophilic graphene oxide IIP using methacrylic acid in the surface imprinting method for Sr²⁺, which is depicted in Figure 8.

Dithiocarbomate (-S₂CNR₂) can form strong metal-carbon bonds. This aspect favors dithiocarbomate-grafted chitosan Sr-IIP preparation. To achieve dithiocarbomate functionalization on chitosan beads, Liu et al. [93] applied the four steps depicted in Figure 9. The important data of Sr-IIPs are presented in Table 4.

### 4.5. Cerium

Cerium is one of the most abundant rare earth metals, and the cerium isotopes with the longest half-life are ¹⁴⁴Ce, ¹³⁹Ce, and ¹⁴¹Ce, produced by uranium fission [127]. Pan et al. [128] prepared chitosan-based Ce^3+^-IIP crosslinked with the γ-3-glycidoxypropyltrimethoxysilane (KH-560) for Ce sorption. Zhang et al. [129] utilized PTW as a base and used a surface imprinting process for PTW and chitosan Ce^3+^ ion-imprinted polymer preparation. Chen et al. [81] developed a carbon paste electrode modified with ion-imprinted polymer for the trace detection of Ce³⁺. Prasad et al. [130] prepared double IIP sensors for the detection of Ce^4+^ and Gd^3+^ ions. The polymer was prepared using free radical polymerization. The magnetic nanoparticle -COCl was spin-coated on the surface of the screen-printed carbon paste electrode after which it was polymerized with multiwalled carbon nanotubes (MWCNT) in the presence of Ce^4+^ and Gd^3+^ template ions (Figure 10).

In another study, Alizadeh et al. [131] prepared a modified carbon electrode with Ce^3+^-IIP ion-imprinted polymer. The monomer used in this 4-vinylpyridine and MAA preparation was polymerized in the presence of Ce(NO_3_)_3_ as a template. Kecili et al. [132] synthesized Ce³⁺-IPP cryogel by using 2-hydroxyethyl methacrylate (HEMA) and *N,N*-methylenebisacrylamide (MBAAm), cryopolymerization, resulting in poly(HEMA-co-(MAAP)₂Ce(H₂O)₂. Liu et al. prepared SBA-15-supported chitosan crosslinked with KH-560 imprinted polymer for Ce^3+^ removal [128]. The adsorption performance and other physical and chemical properties of the abovementioned IIPs are mentioned in Table 5.

### 4.6. Technetium 

Technetium has several different oxidation states in Nature. The oxidation state 7+ is generally the most common in the environment. The pertechnetate ion (TcO₄^−^) is generated in large amounts from nuclear waste, and its mitigation strategy is a key concern [134]. In recent literature, Shu et al. [96] prepared a perrhenate ion-based imprinted polymer by using the surface imprinting method. The perrhenate ion is the structural surrogate and carrier of TcO₄^−^ depicted in Figure 11. The preparation was carried out by applying a surface imprinted strategy, the 4-vinylbenzyl chloride and *N*-vinylimidazole(VI) were polymerized on the surface of vinylated Fe_3_O_4_-SiO_2_. The adsorption capacity was studied using the Langmuir model. The adsorption capacity was found to be equal to 62.8 mg/g at pH = 6 and a temperature of 298.15 K.

### 4.7. Lanthanum

Among rare earth metals, lanthanum is the most abducent one; interest in it has expanded because of its unique physical and chemical properties. Many conventional methods for removing radionuclides and lanthanides from aqueous solutions were already presented [135]. Ion-imprinted materials explicitly and specifically target a metal ion based on their charge and shape. In a recent study, Wang et al. [136] used straw as a base for the preparation of a La³⁺ ion-imprinted polymer using the surface-initiated atom transfer radical polymerization (ATRP) method which is helpful for making homogenous polymer coatings on various surfaces using monomers such as *N,N,N’,N’,N’’*-pentamethyldiethylenetriamine (PMDETA), *N,N*-dimethylaminoethyl methacrylate (DMAEMA), polymerized by thermal radical polymerization and crosslinked using EGDMA. In another study, Mustapa et al. [137] prepared La- and Ce-based IIPs using a Schiff base ligand ((ethyl 4-(2,4-dihydroxybenzylideneamino) benzoate), or an azobenzene ligand ((4-(2,4-dihydroxyphenylazo) acetophenone) and La³⁺ adsorption was studied in the presence of (Pr³⁺, Nd³⁺, and Pm³⁺)—the elements have approximately the same nuclear radius. The prepared IIPs showed promising results for La removal. Besharati-Seidani et al. [138] developed the La³⁺ ion-imprinted nanoparticles (NPs) using the La³⁺-chelating ligand 2,2′:6′,2″-terpyridine (terpy). The adsorption and desorption cycle for this material was completed within 2 to 30 min. Other properties of these IIPs are summarized in Table 6.

### 4.8. Chromium

Recently, for effective binding of Cr ions, a new type of IIPs was developed. Trzonkowska et al. [139] prepared Cr-IIP by using 1,10-phenanthroline ligand. The ligand was polymerized with MAA/ST and crosslinked using EGDMA. The prepared IIPST-AIBN is found to have good adsorption efficiency, rapid adsorption kinetics and desorption, and resistance to competitive ions. In another study, Liang et al. [140] developed surface imprinted Cr-IIP using Fe_3_O_4_-SiO_2_ as a support medium. The polymerization has been done using 4-VP and HEMA or EGDMA as a crosslinker. A similar type of study was carried out by Zhou et al. [141] where Fe_3_O_4_-SiO_2_ was modified with MPS and after surface imprinting polymerization was accomplished by using 4-VP. Kumar et al. [142] used a three-step preparation of chromium-based magnetic IIPs. The first one was the preparation of silica modified Fe_3_O_4_-NPs, in the second step it was modified with amine (Fe_3_O_4_-SiO_2_-NH_2_). Finally, in the third step, the NPs were coated with polymer using 4-VP and MAA, and EGDMA as the crosslinker. Table 7 presents the adsorption performance and other characteristics of Cr-IIPs. 

### 4.9. Nickel

For the selective detection and removal of Ni^2+^, a novel ion-imprinted polymer was developed. Zhou et al. [143] prepared a Ni^2+^-imprinted polymer by bulk polymerization. Ligands such as diphenylcarbazide (DPC) and *N,N*-azobisisobutyronitrile (AIBN) were used for Ni^2+^ adsorption. The study of Ni^2+^ removal from aqueous solutions using surface-modified molecularly imprinted ferrite nanomaterials as adsorbent was carried out by Ahmad et al. [144]. They used ferrous and ferric salts for the development of molecularly imprinted ferrite in a primary medium. A silica coating was applied in the Stober process and the sol-gel method using TEOS and TNT. In another study concerned with the selective Ni removal and proper complexation with Ni ions, an ion-imprinted material was prepared with diacetylmonoxine-modified chitosan. To maintain the structural rigidity of the polymer, glyoxal was used as a crosslinker [145]. Lenoble et al. [146] prepared a Ni^2+^ imprinted polymer using *N*-(4-vinylbenzyl)-2-(aminomethyl)pyridine (Vbamp) ligand to form a complex between Ni^2+^ and the complex crosslinked with EGDMA in a 1:3 ratio. The prepared material showed excellent retention capacity over other competitive ions. The properties of the abovementioned Ni-IIPs are presented in Table 8.

### 4.10. Cobalt

Cobalt is one of the heavy metals that can cause several adverse health effects on humans, e.g., pneumonia, goiter, and gastrological diseases [147,148]. Cobalt has several isotopes, ^60^Co, ^58^Co, ^57^Co that are highly radioactive gamma emitters. Exposure to these isotopes causes hair loss, skin burns, anemia, and cancer [149]. Recently, IIPs have gained significant interest in removing Co^2+^ ions due to their selectivity and recognition ability. The use of inorganic materials (such as palygorskite, titanate whiskers, SBA-15) grafted on chitosan or polyethyleneimine, which have been crosslinked with epoxy functionalized organic or silane coupling agents, for the Co-IIPs preparation, are explored in various studies [150,151,152,153]. Figure 12 presents a method used for the synthesis of IIPs from chitosan.

Nishad et al. [154] prepared Co-IIP from chitosan crosslinked using epichlorohydrin. Bhaskarpillai et al. [155] reported the stability of the complex between cobalt and vinylbenzyliminodiacetic acid (VbIDA) and successfully prepared its Co-IIP. Liu et al. [156] prepared a cysteine -rafted chitosan-based imprinted polymer for Co^2+^ and Mn^2+^ ions. Kang et al. [95] prepared a novel Co-IIP using hydrophilic monomer 1-vinylimdazole (1-VI) surface imprinted on Fe_3_O_4_-SiO_2_ magnetite nanoparticles. The method is depicted in Figure 13.

The use of activated carbon modified with -COOH groups as a crosslinker with hydrazine hydrate for the preparation of Co-IIP was developed by Turan et al. [157]. Metal-organic frameworks (MOF) have interesting properties such as high porosity, a wide range of structural possibilities and tunable porous sizes. Yuan et al. prepared a Co-IIP from MOF based on 2-aminoterephthalic acid (NH_2_-H_2_BDC) modified with glycylglycine [158]. Torkashvand et al. developed a Co-IIP as a voltammetric sensors, aiming for Co^2+^ tracer determination, using Co^2+^-8-hydroxyquinoline linked to cast magnetic nanoparticles at a glass carbon electrode surface. The procedure is illustrated in Figure 14 [159].

In another study, Sebastian et al. [160] developed Co-IIP sensors from multi-walled carbon tubes modified with vinyl groups and crosslinked with N,N’-methylenebis(acrylamide) (NNMBA). Yusof et al. [161] developed dipicolinic acid (DPA)-based Co-IIP by free radical polymerization. Bovine serum albumin (BSA) can go through various conformational and structural changes after reacting with different metal ions via its amino acid residues [162]. Li et al. [163] prepared a Co^2+^-BSA-CO chelated complex. It was introduced as template in a nanocomposite designed from a multi-walled carbon nanotube, Cu nanoparticles and carbon quantum dots-based Co-IIP sensors. Lee et al. [164] used magnetic silica to prepare a Co-IIP with the concern of its main use for nuclear powerplant decontamination. Adibmehr et al. [165] developed novel cobalt ion-imprinted polymer using mesoporous silica SBA-15 modified with (3-chloropropyl)triethoxysilane, and EGDMA as a crosslinker. Biswas et al. [166] prepared curcumin-based ion-imprinted polymer for the selective removal of Co, Cd and Pb ions. The other important parameters of CO-IIPs are presented in Table 9.

## 5. Conclusions

In this review article, ion-imprinted polymers for the adsorption of 10 important radionuclides—U, Th, Cs, Sr, Ce, Tc, La, Cr, Ni, Co—which are found in the nuclear fuel cycle, are presented. These radionuclides were chosen for this study due to their commercial interest and environmental concerns. Recent updates in the field of ion-imprinted polymers for the radionuclide separation discussed with preparation methods, properties, adsorption capacities, and characterization methods used for its determination were discussed here. In recent years, research interest has been focused on the development of ion-imprinted polymers for radionuclide separation due to their properties such as selectivity and recognition ability for targeted ions. Despite the number of studies in this area, more research should be done on TcO₄^−^ and Ce³⁺. Ion-imprinted polymers are based on organic substances that have a huge variability and complex formation abilities with metallic ions whereas inorganic-based materials show excellent thermal and environmental stress-related stability. The combination of both organic- and inorganic-based ion-imprinted polymers allow good preservation of their properties. On the other hand, there are some issues related to ion-imprinted polymers, e.g., regarding their lower reusability and large-scale production. The understanding of the chelating mechanism is important for future research aimed at improving the binding performance of ion-imprinted polymers. Indeed, concerning ligands and functional monomers, this can be achieved by computational methods. Other ways to improve the selective adsorption are by utilizing novel selection methods such as surface imprinting, stimuli-responsive imprinting, and dual/multiple component imprinting. New preparation techniques should be explored for better selectivity, cost reduction, and large-scale production of ion-imprinted polymers.

## Figures and Tables

**Figure 1 materials-14-01083-f001:**
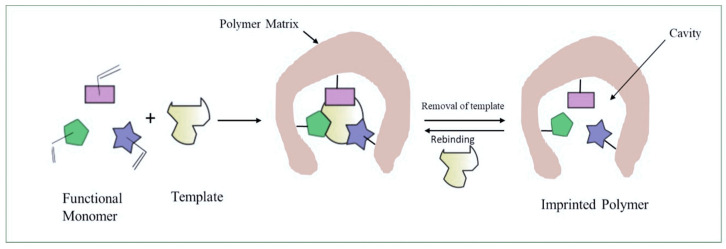
Schematic presentation of the procedure for the preparation of imprinted polymers.

**Figure 2 materials-14-01083-f002:**
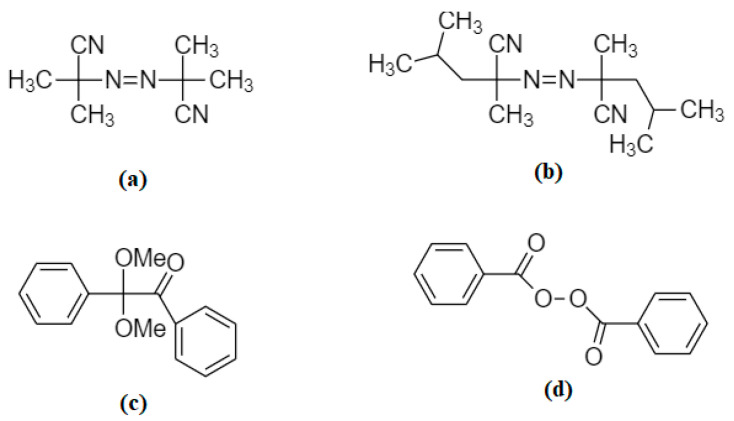
Commonly used initiators: (**a**) azobisisobutyronitrile (AIBN), (**b**) azobisdimethylvaleronitrile (ABDV), (**c**) benzildimethyl acetal, (**d**) benzoyl peroxide (BPO).

**Figure 3 materials-14-01083-f003:**
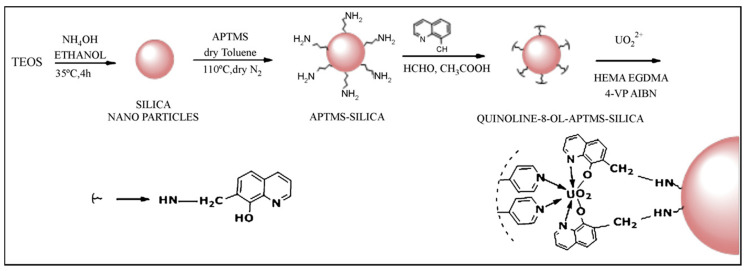
Schematic presentation of uranyl ion-imprinted nanospheres preparation from silica nanoparticles. Reprinted with permission from ref. [100], copyright 2011, Milja et al.

**Figure 4 materials-14-01083-f004:**
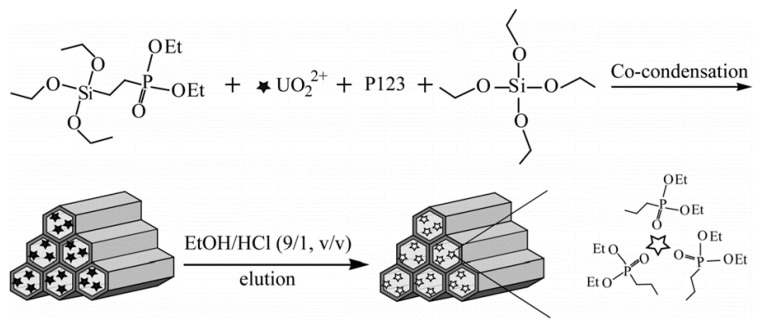
Schematic presentation of the preparation of ion-imprinted mesoporous silica. Reprinted with permission from ref. [103], copyright 2017, Yang et al.

**Figure 5 materials-14-01083-f005:**
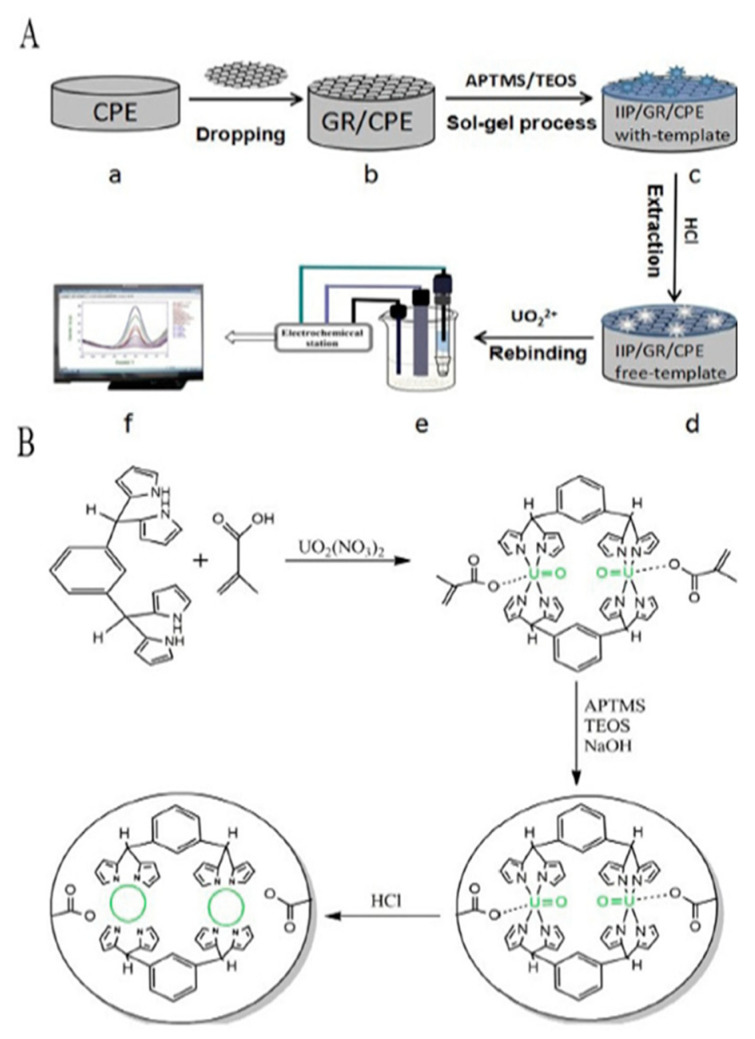
Schematic presentation for the preparation of a uranyl ion-imprinted sensor: (A) surface imprinting polymerization of the U-IIP on the CPE surface (B) the synthesis of U-IIP. Reprinted with permission from ref. [105], copyright 2020, Wang et al.

**Figure 6 materials-14-01083-f006:**
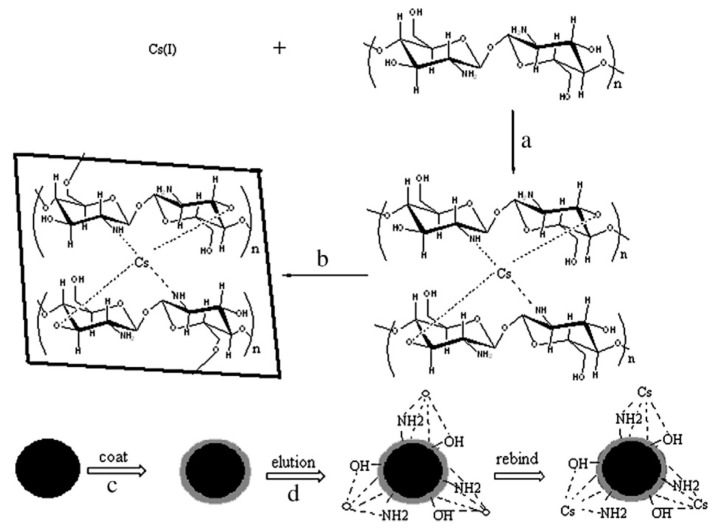
Schematic presentation of the preparation of chitosan Cs^+^ imprinted polymer-coated whiskers. Reprinted with permission from ref. [115], copyright 2020, Othman et al.

**Figure 7 materials-14-01083-f007:**
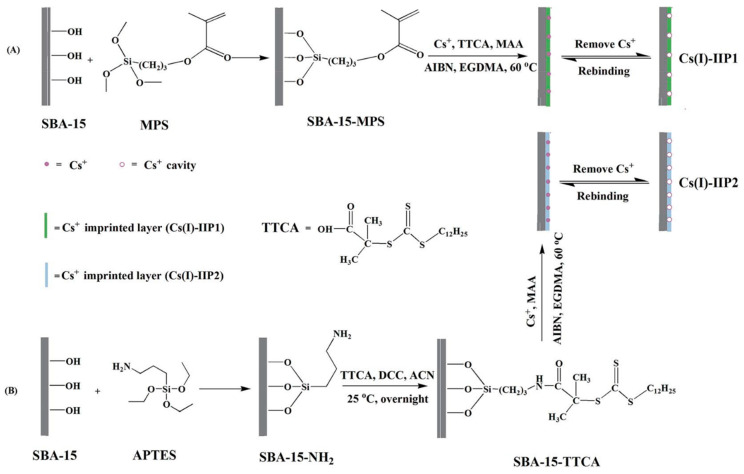
Schematic presentation of two different Cs⁺-IIP, via RAFT polymerization techniques: (A) preparation of Cs(I)-IIP1 (B) preparation of Cs(I)-IIP2. Reprinted with permission from ref. [91], copyright 2015, Meng et al.

**Figure 8 materials-14-01083-f008:**
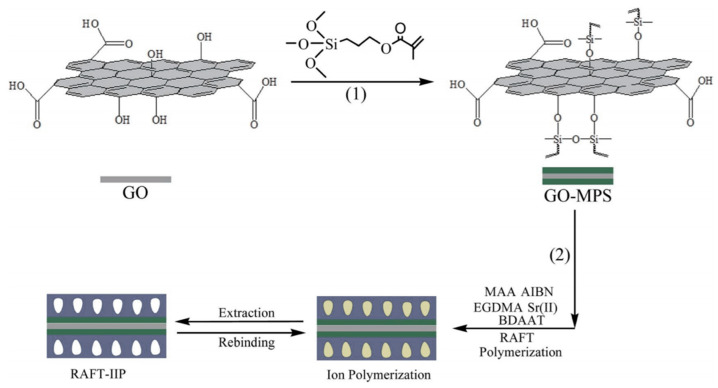
Schematic presentation of graphene oxide-based Sr-IIP. Reprinted with permission from [82], copyright 2015, Liu et al.

**Figure 9 materials-14-01083-f009:**
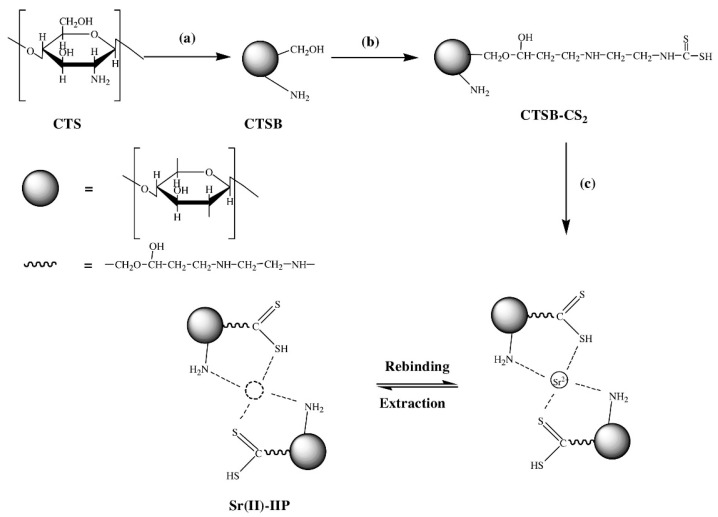
Schematic presentation of the dithiocarbomate functionalized chitosan Sr IIP preparation. Reprinted with permission from ref. [93], copyright 2015, Liu et al.

**Figure 10 materials-14-01083-f010:**
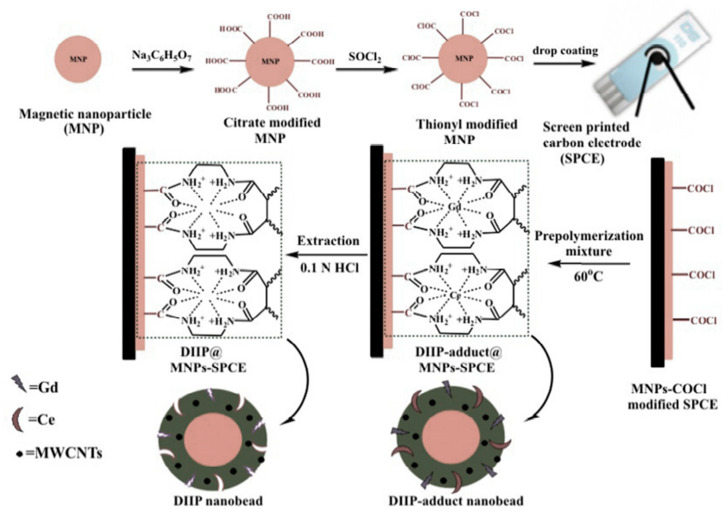
Schematic presentation for the development of the Ce^4+^ and Gd^3+^ ion-imprinted polymer-coated sensors. Reprinted with permission from ref. [130], copyright 2015, Prasad et al.

**Figure 11 materials-14-01083-f011:**
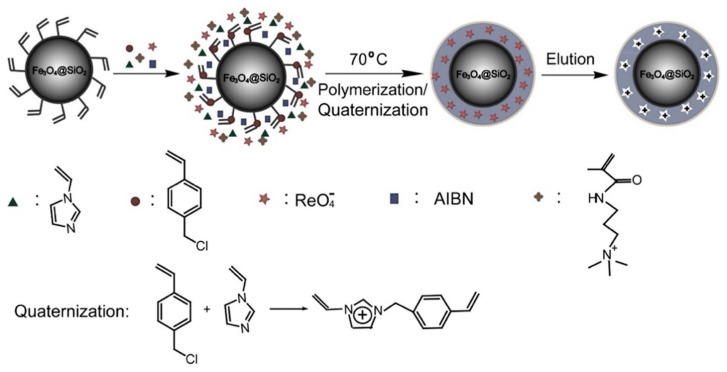
Schematic presentation of pertechnetate IIP preparation. Reprinted with permission from ref. [96], copyright 2015, Shu et al.

**Figure 12 materials-14-01083-f012:**
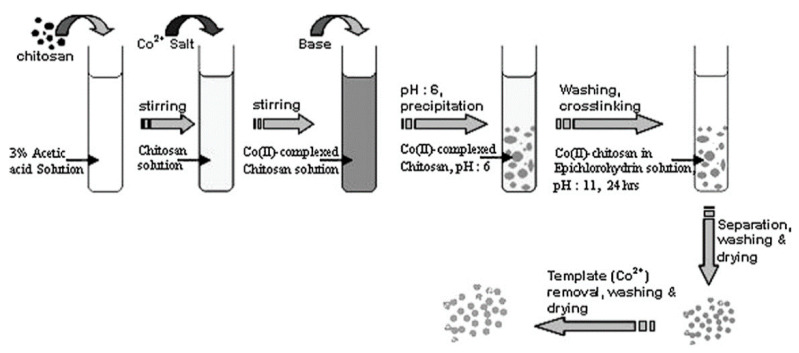
Schematic presentation of Co-IIP prepared from crosslinked chitosan. Reprinted with permission from ref. [154], copyright 2012, Mishad et al.

**Figure 13 materials-14-01083-f013:**
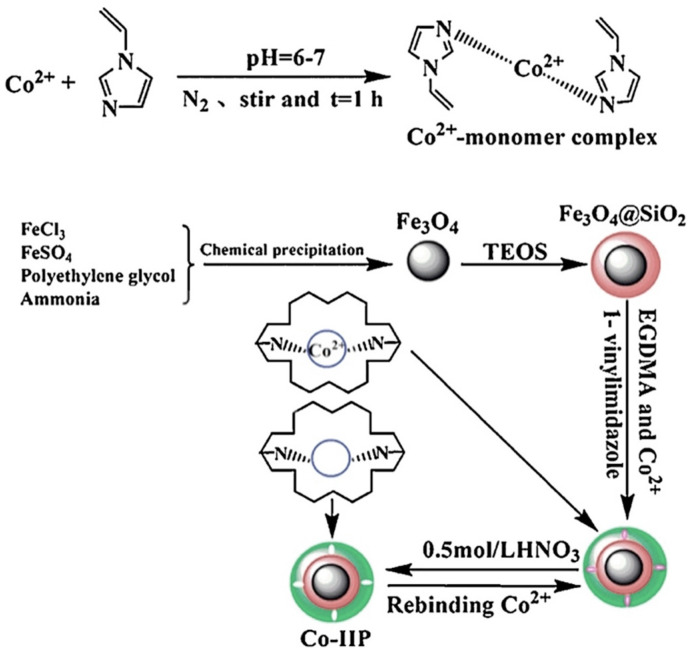
Schematic presentation of the route of preparation of Co-IIP. Reprinted with permission from ref. [95], copyright 2016, Kang et al.

**Figure 14 materials-14-01083-f014:**
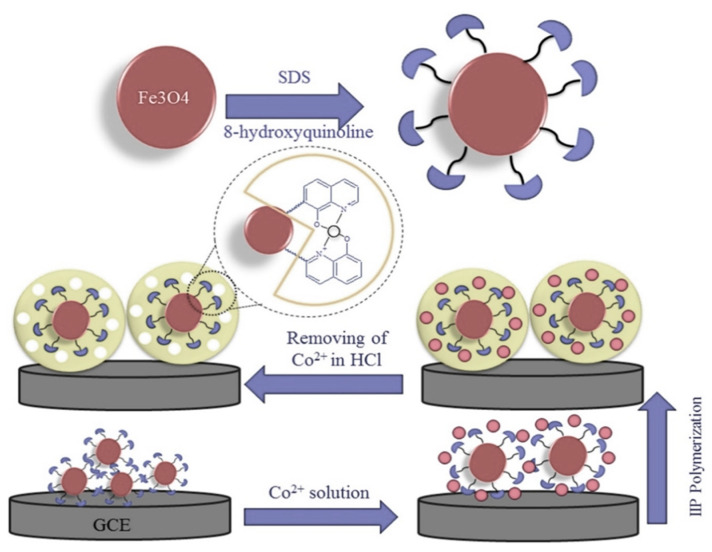
The development of the Co-IIP based voltammetric sensors. Reprinted with permission from ref. [159], copyright 2017, Torkashvand et al.

**Table 1 materials-14-01083-t001:** Uranyl ion-imprinted polymer materials, performance, and characterization.

Substrate	Components	Crosslinker	Method	Adsorption Capacity	pH	Characterization	Ref.
Si-C=C	MAA	EGDMA	FRP	35.92 mg/g	5	FTIR, SEM, EDX.	[60]
-	Na-Alg, Acroylinitrile	Glutarldehyde	Condensation	155 mg/g	5	FTIR, SEM, XRD, XPS	[72]
CMC	Acrylonitrile, SAL, thiourea	Formaldehyde	FRP	180 mg/g	5	FTIR, TGA, SEM	[79]
Fe_3_O_4_	MAA, SALO, 4-VP, Oleic acid	EGDMA	FRP	1.01 mg/g	4	FT-IR, Raman, SEM	[90]
Fe_3_O_4_	APS and TEOS	-	Sol-gel	25.8 µmol/g	4	FTIR, XRD, SEM	[92]
Fe_3_O_4_-SiO_2_	HEMAA, VI	-	FRP	146.4 mg/g	5	FTIR, SEM, EDX.	[96]
-	(pyr_2_en)DMSO 4-VP	Styrene	FRP	-	7	UV-VIS, IR, XRD	[99]
HQ-APTMS-SI	4-VP, HEMA	EGDMA	FRP	97.1 µmol/g	6	FTIR, TEM, SEM	[100]
-	Chitosan and PVA	EGDMA	Condensation	132 mg/g	5–6	FTIR	[101]
Fe_3_O_4_	Chitosan	Glutaraldehyde	Crosslinking	187.3 mmol/g	5	FTIR, XRD, SEM	[102]
-	DPTES, P123, TEOS	-	Condensation	80 mg/g	5.9	FTIR, SEM, TEM, ³¹P NMR, TGA	[103]
HPC	-	-	Carbonization	503.64 mg/g	8	SEM, XRD, XPS, FTIR	[104]
GR/CPE	IPTP, APTES, TEOS	-	Sol-gel	-	5	FTIR, EDS, SEM	[105]
-	Chitosan	-	Hydrothermal	408.2 mg/g	7	FTIR, EDS, SEM	[106]
-	Methacrylic acid and triethylamine	EGDMA	FRP	23.9 mg/g		FTIR, TGA	[107]
γ-MPS-Fe_3_O_4_	SALO, 4-VP, MAA	EGDMA	FRP	-	4.3	FTIR, SEM	[108]
-	Salicyaldehyde, p-aminostyrene,	DVB	FRP	147.8 mg/g	5	¹H and ¹³C NMR, FT-IR, SEM	[109]
Carbon powder	AQ, ICTMS	-	Sol-gel	-	5	SEM	[110]

**Table 2 materials-14-01083-t002:** Thorium ion-imprinted polymer materials, performance, and characterization.

Substrate	Components	Crosslinker	Method	Adsorption Capacity	pH	Characterization	Ref.
SiO_2_-modified MA	MAA	EGDMA	FRP	7.3 mg/g	3	FTIR, UV-VIS	[78]
SiO_2_-NH_2_	DBM, β-CD	EGDMA	FRP	30.8 mg/g	3.5	FT-IR, XPS, SEM	[80]
-	PMTCAAP	EGDMA	FRP	64.8 mg/g	3.5	UV-VIS, FTIR, ¹H NMR	[112]
Vinyl functionalized Fe_3_O_4_ SiO_2_-NH_2_	BASPDA	EGDMA	FRP	42.54 mg/g	4.5	FTIR, XRD	[113]
Fe_3_O_4_	Chitosan	ECH	Condensation	147.7 mg/g	4	FTIR, EDX	[114]
-	2-HMPAD, 2-HMPAA	EGDMADVB	FRP	-	3.5	FTIR, SEM, XPS.	[115]

**Table 3 materials-14-01083-t003:** Cesium ion-imprinted polymer materials, performance, and characterization.

Substrate	Components	Crosslinker	Method	Adsorption Capacity	pH	Characterization	Ref.
SBA-15	MAA	EGDMA	RAFT	54.54 mg/g	6	FTIR, SEM, EDS	[91]
STW	Chitosan	KH-560	Condensation	32.88 mg/g	6	FTIR, XRD	[118]
-	DB24C8, MAA	EGDMA	FRP	-	9	SEM	[119]
-	PAN	-	Spin casting	-	-	SPR	[120]

**Table 4 materials-14-01083-t004:** Strontium ion-imprinted polymer materials, performance, and characterization.

Substrate	Components	Crosslinker	Method	Adsorption Capacity	pH	Characterization	Ref.
-	TSPA	-	Sol-gel	-	7	-	[11]
Graphite	MAA, MPS	EGDMA	RAFT	145.8 mg/g	6	FTIR, UV-VIS, SEM, TEM, XRD, AFM	[82]
-	Chitosan, CS_2_	Glutaraldehyde	Condensation	86.66 mg/g	6	FTIR, XPS, ICP, SEM, FAAS	[93]
Yeast	TEOS, chitosan	KH-560	Condensation	60.6 mg/g	6	FTIR, SEM, SEM, TEM	[94]
-	D18C6, MAA	EGDMA	FRP	-	6.3	FTIR, LSA	[123]
Palygorskite	Chitosan	KH-560	Condensation	45 mg/g	4	FTIR, SEM	[124]
Fe_3_O_4_-SBA-15	MPS, MAA, NIPAM	BIS	FRP	89 mg/g	7	FTIR, XRD, TGA, SEM	[125]
SBA-15	MAA, TTCA	EGDMA	FRP	22.12 mg/g	6	FTIR, SEM, TEM	[126]

**Table 5 materials-14-01083-t005:** Cerium ion-imprinted polymer materials, performance, and characterization.

Substrate	Components	Crosslinker	Method	Adsorption Capacity	pH	Characterization	Ref.
Graphite	APA	EGDMA	FRP	-	-	-	[81]
PTW	Chitosan	KH-560	Condensation	-	6	FTIR, XRD,	[129]
MNPs, SPCE	-	EGDMA	FRP	-	-	SEM	[130]
MWCN	MAA, VP	DVB	FRP	-	-	SEM	[131]
-	HEMA, MBAAm	-	Cryolpoymerisation	-	6–7	FTIR, SEM, EDX, UV	[132]
SBA-15	Chitosan	KH-560	Condensation	-	5	FTIR, XRD, SEM	[133]

**Table 6 materials-14-01083-t006:** Lanthanum ion-imprinted polymer materials, performance, and characterization.

Substrate	Components	Crosslinker	Method	Adsorption Capacity	pH	Characterization	Ref.
Straw	PMDETA, DMAEMA	EGDMA	ATRP	125 mg/g	6	XPS, SEM, FTIR	[136]
-	Schiff base, and Azobenzene	EGDMA	FRP	25 mg/g 24 mg/g	7	FTIR, FE-SEM, UV-VIS	[137]
	Terpy	EGDMA	FRP	133.8 mg/g	3.5	FTIR, UV, SEM.	[138]

**Table 7 materials-14-01083-t007:** Chromium ion-imprinted polymer materials, performance, and characterization.

Substrate	Components	Crosslinker	Method	Adsorption Capacity	pH	Characterization	Ref.
-	1,10-phenanthroline complex, styrene, MAA	EGDMA	Bulk polymerization	-	4.5	FTIR, SEM	[139]
Fe_3_O_4_-SiO_2_	4-VP, HEMA	EGDMA	FRP	311.95 mg/g	2	FTIR, SEM-EDS, XRD, TGA	[140]
Fe_3_O_4_-SiO_2_	4-VP	EGDMA	FRP	131.40 mg/g	2	FTIR, XPS, TEM, TGA	[141]
Fe_3_O_4_-SiO_2_-NH_2_	4-VP, MAA	EGDMA	FRP	169.49 mg/g	2	FTIT, SEM-EDS	[142]

**Table 8 materials-14-01083-t008:** Nickel ion-imprinted polymer materials, performance, and characterization.

Substrate	Components	Crosslinker	Method	Adsorption Capacity	pH	Characterization	Ref.
	DPC, MAA	EGDMA	FRP	86.3 mg/g	7	FTIR, SEM, XRD, EDX	[143]
Fe_3_O_4_-SiO_2_	TNT, TEOS	-	Sol-gel	2.64 mg/g	7.6	FTIR, SEM, EDX, XRD	[144]
diacetylmonoxine modified chitosan	-	glyoxal	Condensation	135 mg/g	5	FTIR, XRD, ¹H NMR, ¹³C NMR, SEM, EDX	[145]
-	Vbamp	EGDMA	FRP		4–7	FTIR, UV-VIS, SEM	[146]

**Table 9 materials-14-01083-t009:** Cobalt ion-imprinted polymer materials, performance, and characterization.

Substrate	Components	Crosslinker	Method	Adsorption Capacity	pH	Characterization	Ref.
Fe_3_O_4_-SiO_2_	1-VI	EGDMA	FRP	23.09 mg/g	6–7	FTIR, SEM	[95]
Palygorskite	Chitosan	Condensation	31.5 mg/g		4	FTIR, SEM	[151]
SBA-15	PEI	ECH	Condensation	39.36 mg/g	7	FTIR, SEM, XRD	[152]
STW	Chitosan	KH-560	Condensation	33.7 mg/g	5	FTIR, SEM	[153]
-	Chitosan	ECH	Condensation	75 µmol/g	4.8	-	[154]
-	Cu(VbIDA)	EDGMA	FRP	205 µmol/g	5	EPR	[155]
-	Cytesine, chitosan	ECH	Condensation	-	5.58	FTIR, SEM	[156]
AC-COOH	Hydrazine hydrate	-	condensation	833.7 mg/g	7	FTIR, SEM	[157]
MOF	NH_2_-H_2_BDC, ZrCl_4,_ glycine	-	-	175 mg/g	8.4	FTIR, SEM, XRD	[158]
8-HQ modified Fe_3_O_4_	AAM,	NNMBA	FRP	-	9	FTIR, SEM	[159]
Vinyl functionalized MWCNT	Acrylic acid	NNMBA	FRP	-	6	FTIR, SEM, EDAX, TGA	[160]
-	MAA,4-VP	EGDMA	FRP	106 mg/g	6	FTIR, SEM	[161]
MWCNT	BSA, Carbon dots, PVP	-	-	-	7.8	XPS, SEM	[163]
Fe_3_O_4_-SBA-15	TEOS, P123	-	Sol-gel	74 mg/g	8	FTIR, SEM, TGA	[165]
-	Curcumin, acrylic acid	EGDMA	FRP	105 mg/g	6	UV-VIS, FTIR, SEM	[166]

## Data Availability

Data is contained within the article.

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
