# Peer review of "Ion-Imprinted Polymers: Synthesis, Characterization, and Adsorption of Radionuclides"

_materials, 2021, doi:10.3390/ma14051083_

Round 1

Reviewer 1 Report

See attachment

Author Response

Dear reviewer,

first let me say thank you for our manuscript reviewing and consideration, for the comments and suggestions. We revised the enclosed manuscript by your suggestions and prepared it in a more presentable and understandable way for readers.

Reviewer #1

Comment 1: There are too many English mistakes and many scientific confusions.

Response 1: Thank you very much for reviewing the manuscript and for the suggestions. We apologies there were English mistakes in the manuscript. We tried to correct and minimize all the mistakes by checking it by native speaker. By your suggestion, we revised the whole manuscript and prepared it in a more presentable and understandable way for readers.

Comment 2: There is confusion between MIP and IIP. It is not clear about the differences. For instance, a covalent method is used in another sense in IIPs than in MIPs

Response 2: Thank you for the comment. The main article’s aim is focused onto the IIP, whereas MIP is a more common or broader term used in the sense of imprinting technology. We apologies for mixing those terms. In the article, we made the necessary changes (removing the term MIP used IIP) to reduce this confusion

i.e., P. 3; L: 163, 164, 203

Comment 3: The synthesis methods are also mixed: Free radical polymerization, then sol-gel, then again radical polymerization. Etc. A better systematization is needed.

Response 3: Thank you for the comment. By your suggestion, we revised the whole manuscript and prepared it in a more presentable and understandable way for readers.

Comment 4: The role of the ligands is not clearly explained.

Response 4: Thank you for the comment. The ligand role in IIPs for the removal of radionuclides is better explained in the revised manuscript

  1. 5, L: 308-342

Comment 5: What is CPE? But GR/CPE in figure 5?

Response 5: Thank you for the comment. We apologies for miss-understanding with the abbreviation part. All necessary changes have been made in the whole manuscript. The separate documents with all abbreviation are in supplementary materials.

P.10; L: 704

Comment 6: What are the abbreviations in Table 1? Also, some abbreviation in Table 2? And Table?

Response 6: Thank you for the comment. The abbreviations are provided in a separate document and it is mentioned in the article when they appear for the first time.

Comment 7: What is PTW? What is GPTMS silane etc.? All abbreviation in the article!!

Response 7: Thank you for the comment. The abbreviations are provided in a separate document (supplementary material) and it is mentioned in the article when they appear for the first time

i.e. P. 9; L:718

Comment 8: What do you mean by:

  1. Li et al. [10] has prepared strontium ion-imprinted hybrid gel from bis(trimethoxysilylpropyl) amine (TSPA) for the removal Sr² and Ca² dicyclohexano-18-crown-6, methacrylic acid (MAA), EGDMA removal?

Response 8: We apologies for the error and mistakes during writing this part. The sentence and all manuscript are revised

  1. 9; L: 721

Reviewer 2 Report

The manuscript is well prepared, written in the correct scientific language. It concerns the synthesis and adsorption properties of ion-imprinted polymers that can be applied to remove radionuclides. The usage of selected adsorbents can reduce the negative impact of radioactive waste on the environment. This issue is current and important from the point of view of nature protection. The publication correctly summarizes various, numerous studies from recent years. The authors also emphasize what should still be investigated in order to complete the data on selected adsorbents. Therefore, I recommend acceptance of the manuscript for publication.

Author Response

Dear reviewer,

thank you for our manuscript reviewing and consideration, for the comments and suggestions. 

Authors

Reviewer 3 Report

The review concerns synthesis, characterization, and application of ion-imprinted polymers as adsorbents of radionuclides. The paper is well written but some details should be added.

  1. The adsorption process onto different adsorbents is complex. There is a lot of parameters that determine its efficiency. In the review, I cannot find information about the influence of pH – which is important for adsorption, because it determines the form in which ions occur in the water systems. There should be also information about the adsorbent behavior in the reaction system with different pH. What about the influence of adsorbent dosage and adsorbate concentration? Why these parameters weren’t analyzed? In order to optimize the adsorption process detailed description of the influence of each of those parameters should be presented.
  2. I think that is not necessary to show in the review formula of Langmuir and Freundlich isotherms. It is better to analyze what kind of isotherm was observed in the case of the adsorption of radionuclides in the literature.
  3. The “Conclusion” section should be rewritten to reflect the findings and content of the manuscript.

Author Response

Dear reviewer,

First let me say thank you for our manuscript reviewing and consideration, for the comments and suggestions. By the all yours suggestions, we revised the enclosed manuscript and prepared it in a more presentable and understandable way for readers.

Authors

Reviewer #3

Comment 1: The adsorption process onto different adsorbents is complex. There is a lot of parameters that determine its efficiency. In the review, I cannot find information about the influence of pH – which is important for adsorption, because it determines the form in which ions occur in the water systems. There should be also information about the adsorbent behavior in the reaction system with different pH. What about the influence of adsorbent dosage and adsorbate concentration? Why these parameters weren’t analyzed? In order to optimize the adsorption process detailed description of the influence of each of those parameters should be presented.

Response 1: Thank you very much for this comment the valuable instruction regarding revising the manuscript. The parameters mentioned in your comment are essential for describing the properties of the ion-imprinted polymers, mainly with focus for the radionuclide’s removal. We included the sections on the pH influence on the adsorption process and the influence of adsorption dosage and adsorbate concentration in the manuscript.

P.7; L: 505-532

Comment 2: I think that is not necessary to show in the review formula of Langmuir and Freundlich isotherms. It is better to analyze what kind of isotherm was observed in the case of the adsorption of radionuclides in the literature.

Response 2: Thank you very much for the suggestion. The isotherm part of the manuscript was revised by your suggestion. The formulas remain in manuscript, but we put more information about the isotherm mostly used for adsorption process description.

P.8; L: 432-504

Comment 3: The “Conclusion” section should be rewritten to reflect the findings and content of the manuscript.

Response 3: Thank you very much for the suggestion. The conclusion part was rewritten which will reflect the findings and the content of the manuscript.

Reviewer 4 Report

-The English of the manuscript is weak and must be improved. For example, in this sentence (lines 64-65): “ The needing’s for RN separation from aqueous solutions are all the greater, the longer their half-life >200 000 years: ⁷⁹Se, ⁹³Zr, ⁹⁹Tc, ¹⁰⁷Pd, ¹²⁶Sn, ¹²⁹I, ¹³⁵Cs is.” The grammatical structure is wrong and the sentence has no meaning. There are many more examples. Please ask a native English speaker to proofread your manuscript

-Lines 28-103: Comprehensive information for radioactive materials has been provided which is not necessary. All this information make your manuscript long and boring for readers. You can simply mention the important topics and provide suitable references. Instead, section 2 can be embedded into the introduction section. This review must be focused on IIPs and their application for radionuclides.  

-Line 136: molecules must be deleted. IIPs are for ions and MIPs for molecules. However, it could be mentioned that an IIP is synthesized for a negatively- or positively-charged molecule.

- Lines 137-138: This sentence has no meaning: “IIPs were created based on the reaction of the enzyme and the substrate,”. It must be removed or corrected.

-Lines 139-140: “Imprinting of the template molecules in a polymer matrix…”, In this review IIP and ions are the topic and not MIP and molecules.

-Lines 145-146: “…which are based on the reaction of enzymes and antibodies.”, it is not based on.., but it imitate that.

-Line 150-151: “The MIPs preparation …the molecules/ions.” Please be careful MIPs are for molecules, and IIPs for ions.

-Line 156: Is the term “secondary bonds” correct to be used here? Which types of interactions are classified as secondary bonds? Is ionic bond categorized as a secondary bond?

- Lines 171-172: This is wrong: “the functional groups containing monomers”. Monomers containing functional groups is correct.

-Line 174: “template molecules“, for IIPs template are ions.

-Line 174-175: “generating specific binding sites and pores”, removing the templates produces the recognition sites. Creation of pores needs other synthetic strategies.

-Throughout the manuscript: please define abbreviations and then use it. E.g. lines 203-205: “free radical polymerization” is mentioned in line 203 and then in lines 204-205 is mentioned “free radical polymerization (FRP)”.

-Lines 248-249: “The simplest method for the IIPs synthesis generation is free radical polymerization. This method is attractive due to control over morphology.” FRP is a method which start and propagate very fast which is generally out of control. It is weird that you wrote it is attractive due to control over morphology. Please provide suitable references that confirm your claim. To control the polymerization, new synthesis method is used like controlled living radical polymerization (RAFT is one type of this polymerization) ….

-Line 252: What is “solution [80]”?

-Lines 265-267: non imprinted polymer is not describe well. You can simply write, NIP is synthesized using the same procedure but without the template ions. The sentences must be rewritten.

-There some sections which must be removed. They are obvious and do not need to be mentioned and discussed. For example ” 8. Influence of the adsorbent dosage and adsorbate concentration”. It is clear, when you increase the amount of adsorbent, the amount of adsorbed analyte will be increased due to the enhancement of active sites…

-An important question: The title of the manuscript is: “Ion-imprinted polymers: synthesis, characterization, and adsorption of radionuclides”. In introduction section, it is also emphasized that this review will focus on radionuclides. In section “9. Application of ion-imprinted polymers for radionuclides”, are all of the mentioned manuscripts evaluated the radionuclides? If not, they must be removed or the title and description of the review must be changed.

Author Response

REVIEW

Dear reviewer,

First let me say thank you for our manuscript reviewing and consideration, for the comments and suggestions. We appreciate the efforts that have been taken by you and the reviewer. By all-reviewer’s suggestions, we revised the enclosed manuscript.

Here is a point -by-point response to the reviewer’s comments and concerns.

Reviewer #4

Comment 1: The English of the manuscript is weak and must be improved. For example, in this sentence (lines 64-65): “ The needing’s for RN separation from aqueous solutions are all the greater, the longer their half-life >200 000 years: ⁷⁹Se, ³Zr, ⁹⁹Tc, ¹⁰⁷Pd, ¹²Sn, ¹²I, ¹³Cs is.” The grammatical structure is wrong and the sentence has no meaning. There are many more examples. Please ask a native English speaker to proofread your manuscript.

Response 1: Thank you for comment. The manuscript was checked by native speaker.

Comment 2: Lines 28-103: Comprehensive information for radioactive materials has been provided which is not necessary. All this information make your manuscript long and boring for readers. You can simply mention the important topics and provide suitable references. Instead, section 2 can be embedded into the introduction section. This review must be focused on IIPs and their application for radionuclides.

Response 2: Thank you for comment. The sections of manuscript were revised, lot of information were added to manuscript via review process as suggestion of reviewers.

Comment 3: Line 136: molecules must be deleted. IIPs are for ions and MIPs for molecules. However, it could be mentioned that an IIP is synthesized for a negatively- or positively-charged molecule.

Response 3: Thank you for the comment. The change has been made by your recommendation and the sentence has also been reconstructed.

Comment 4: Lines 137-138: This sentence has no meaning: “IIPs were created based on the reaction of the enzyme and the substrate,”. It must be removed or corrected.

Response 4: Thank you for comment, we apologize for this mistake. The sentence has been removed from the manuscript.

Comment 5: Lines 139-140: “Imprinting of the template molecules in a polymer matrix…”, In this review IIP and ions are the topic and not MIP and molecules.

Response 5: Thank you for pointing out this. The appropriate change has been made in the sentence. P.3, L: 136

Comment 6: Lines 145-146: “…which are based on the reaction of enzymes and antibodies.”, it is not based on.., but it imitate that.,

Response 6: Thank you for the comment. Suggested change has been made in the sentence. P.3, L: 141-142

Comment 7: Line 150-151: “The MIPs preparation …the molecules/ions.” Please be careful MIPs are for molecules, and IIPs for ions.

Response 7: Thank you for the comment. The appropriate change has been made P.4, L: 241-242

Comment 8:  Line 156: Is the term “secondary bonds” correct to be used here? Which types of interactions are classified as secondary bonds? Is ionic bond categorized as a secondary bond?

Response 8: We apologies for using the term secondary bonds. The correct term non-covalent bonds have been used in the sentence. However, Non-covalent bonds are also can be secondary bonds (Hydrogen bonds, Vander Waal’s interaction, hydrophobic interaction) which are the weak attractive forces between electric dipoles between the molecules or atoms. The ionic bonds cannot be categorized as secondary bonds because they don’t have dipole-dipole interaction instead of that it involves electrostatic interaction between oppositely charged ions. P.4, L: 244-247

Comment 9: Lines 171-172: This is wrong: “the functional groups containing monomers”. Monomers containing functional groups is correct.

Response 9: Thank you for this suggestion. We apologies for the wrong sentence, the sentence has been corrected. P.4, L: 224-225

Comment 10: Line 174: “template molecules“, for IIPs template are ions.

Response 10: Thank you for the comment. Suggested change has been made. P.4, L: 227

Comment 11: Line 174-175: “generating specific binding sites and pores”, removing the templates produces the recognition sites. Creation of pores needs other synthetic strategies.

Response 11: Thank you for the comment. The mentioned sentence was corrected.

Comment 12: Throughout the manuscript: please define abbreviations and then use it. E.g. lines 203-205: “free radical polymerization” is mentioned in line 203 and then in lines, 204-205 is mentioned “free radical polymerization (FRP)”.

Response 12: Thank you for the comment. The abbreviations in the whole manuscript were checked again. Also please find the list of abbreviation in the supplementary materials.

Comment 13: Lines 248-249: “The simplest method for the IIPs synthesis generation is free radical polymerization. This method is attractive due to control over morphology.” FRP is a method which start and propagate very fast which is generally out of control. It is weird that you wrote it is attractive due to control over morphology. Please provide suitable references that confirm your claim. To control the polymerization, new synthesis method is used like controlled living radical polymerization (RAFT is one type of this polymerization) ….

Response 13: We agree with the comment you made and thank you for pointing out this error. The sentence has been revised. P. 6-7, L: 472-489

Comment 14: Line 252: What is “solution [80]”?

Response 14: The “solution [80]” is the solution polymerization method. These terms are properly mentioned in the sentence as i.e. solution polymerization. P. 6; L: 472-473

Comment 15: Lines 265-267: non imprinted polymer is not describe well. You can simply write, NIP is synthesized using the same procedure but without the template ions. The sentences must be rewritten.

Response 15: Thank you for pointing out this. We defined the NIP as you suggested in that section. P. 7, L: 503-504

Comment 16: There some sections which must be removed. They are obvious and do not need to be mentioned and discussed. For example ” 8. Influence of the adsorbent dosage and adsorbate concentration”. It is clear, when you increase the amount of adsorbent, the amount of adsorbed analyte will be increased due to the enhancement of active sites…

Response 16: Thank you very much for you comment. The section of manuscript, you have mentioned in your comments, was added via review process, because it was strongly suggested by another reviewer and we decided to leave it in this way.

Comment 17: An important question: The title of the manuscript is: “Ion-imprinted polymers: synthesis, characterization, and adsorption of radionuclides”. In introduction section, it is also emphasized that this review will focus on radionuclides. In section “9. Application of ion-imprinted polymers for radionuclides”, are all of the mentioned manuscripts evaluated the radionuclides? If not, they must be removed or the title and description of the review must be changed.

Response 17: According to the title and main aim of this review article the whole manuscript is focused onto the adsorption of ten important radionuclides – U, Th, Cs, Sr, Ce, Tc, La, Cr, Ni, Co, by IIPs and its fundamental aspects. In the section: introduction, we mentioned basics about these 10 RNs, which are found in the nuclear fuel cycle. In the section “5 Application of IIPs for radionuclides removal (previous section n.9) we described these 10 RNs with relation to IIPs. In the case that any the manuscript used in this section is not focused onto the specific radionuclide, each of it evaluated at least its analogue

Round 2

Reviewer 1 Report

It is a very bad article. You did not understood the difference between MIPs and IIPs. 

You improved the English, but, there are a lot of poor English phrase.

For instance:

Although, the adsorbents used in nuclear waste management improved over time in their structural, physical, and chemical aspects [36]. The materials made for adsorption purposes having drawbacks, such as a lack of selectivity towards the targeted compounds or elements.

Highly selective monitoring of metal ion by imprinted materials by Hande et al. [39], recent advances in imprinted polymers by Branger et al. [40], the review on imprinted polymer developed from the halloysite nanotubes for environmental pollutant removal [41] and more [42, 43][41, 42

Thus, IIPs offer more excellent selectivity and affinity towards analytes and economically feasible

I stop here

You still are mixing IIP with MIPS. See below:

Ion- iImprinted polymersIIPs (MIPs) are the types of materials developed for extraction, filtration, and purification of a targeted ions/molecules from the solid and liquid media [44]. These materials  are generally crosslinked polymers with pores and the binding site for the targeted ions or molecules

The part below is about MIPs, but not about IIPs:

Two approaches can prepare IIPs, one is covalent interaction, and the other one is noncovalent  interaction. The covalent method of molecularly imprinting technique developed by Wulff et al. [47] IIPs can be prepared by two approaches one is covalent interaction and the other is noncovalent interaction. The covalent method of molecularly imprinting technique developed by Wulff et al. [48] which involves reversible covalent binding between monomers and the templates, this approach  provides greater stability and consistent recognitions sites compared to the non-covalent approach. In the non-covalent method secondary bonds (hydrogen bond, ionic bond, etc.) are used by monomers to self-assemble around the template after polymerization, the template is removed from  the material. Mosbach et al. [48] developed this method, which is an effective form of cavities that  consist of memory of the template’s shape and charge: responsible for the selective recognition of the analytes. The non-covalent method is easy to use and has the option to choose from a range of different monomers.

For IIPs there are big differences in comparison with MIPs: you have also a covalent method, but in which a monomer containing the active ligand and the ion is copolymerized with the crosslinking monomer, and a noncovalent method, in which a complexation compound, containing the ligand and the ion is self assembled with a functional monomer and after the crosslinkink it embedded in the polymer matrix. The extraction does not remove the ligand but only the ion!

So that point 3 Basic is not correct also, becuase you should renmove only the ion, which is binded by ligand. So 3.2.1 is not correct too.

I stop here! There are enough arguments to reject the article.

Author Response

Dear reviewer,

please in attachment finds the answers on you comments.

Authors

Reviewer 4 Report

The manuscript has improved and can be published in the present form.